# Physiopathology of the Permeability Transition Pore: Molecular Mechanisms in Human Pathology

**DOI:** 10.3390/biom10070998

**Published:** 2020-07-04

**Authors:** Massimo Bonora, Simone Patergnani, Daniela Ramaccini, Giampaolo Morciano, Gaia Pedriali, Asrat Endrias Kahsay, Esmaa Bouhamida, Carlotta Giorgi, Mariusz R. Wieckowski, Paolo Pinton

**Affiliations:** 1Department of Medical Sciences, Laboratory for Technologies of Advanced Therapies (LTTA), University of Ferrara, 44121 Ferrara, Italy; simone.patergnani@unife.it (S.P.); rmcdnl@unife.it (D.R.); mrcgpl@unife.it (G.M.); pdrgai@unife.it (G.P.); asratendrias.kahsay@unife.it (A.E.K.); bhmsme@unife.it (E.B.); grgclt@unife.it (C.G.); 2Maria Cecilia Hospital, GVM Care & Research, Via Corriera 1, Cotignola, 48033 Ravenna, Italy; 3Laboratory of Mitochondrial Biology and Metabolism, Nencki Institute of Experimental Biology of the Polish Academy of Sciences, 3 Pasteur Str., 02-093 Warsaw, Poland; m.wieckowski@nencki.edu.pl

**Keywords:** mitochondrial permeability transition, apoptosis, necrosis, ischemia/reperfusion, cancer, neurodegeneration, cyclosporin A

## Abstract

Mitochondrial permeability transition (MPT) is the sudden loss in the permeability of the inner mitochondrial membrane (IMM) to low-molecular-weight solutes. Due to osmotic forces, MPT is paralleled by a massive influx of water into the mitochondrial matrix, eventually leading to the structural collapse of the organelle. Thus, MPT can initiate outer-mitochondrial-membrane permeabilization (MOMP), promoting the activation of the apoptotic caspase cascade and caspase-independent cell-death mechanisms. The induction of MPT is mostly dependent on mitochondrial reactive oxygen species (ROS) and Ca^2+^, but is also dependent on the metabolic stage of the affected cell and signaling events. Therefore, since its discovery in the late 1970s, the role of MPT in human pathology has been heavily investigated. Here, we summarize the most significant findings corroborating a role for MPT in the etiology of a spectrum of human diseases, including diseases characterized by acute or chronic loss of adult cells and those characterized by neoplastic initiation.

## 1. Introduction

Mitochondrial permeability transition (MPT) remains one of the most unusual and poorly characterized aspects of mitochondrial biology. This phenomenon was first reported in the late 1970s and was originally considered an artefact due to the experimental conditions required to investigate isolated mitochondria. However, (as better described below) MPT is triggered by the accumulation of Ca^2+^ in the mitochondrial matrix that occurs in isolated mitochondria exposed to a [Ca^2+^] significantly higher than that in the cytoplasm. The development of techniques to measure Ca^2+^ within different compartments later demonstrated that mitochondria can actively uptake Ca^2+^ even in living cells, prompting the investigation of MPT in cell pathophysiology.

### 1.1. Mitochondrial Routes of Cell Death

Mitochondria actively participate in multiple forms of regulated cell death (RCD) through different routes, all involving major alterations in the outer mitochondrial membrane (OMM) and/or inner mitochondrial membrane (IMM). Activation of the mitochondrial pathway of RCA causes the redistribution of mitochondrial proteins into the cytoplasm, which activates cell-death effectors, or the dramatic impairment of cell bioenergetics, which ultimately leads to death of the affected cells. These mechanisms are, in general, categorized into two types that function under different conditions: those involving only outer-mitochondrial-membrane permeabilization (MOMP) and those in which there is also a long-lasting increase in IMM permeability, the MPT. MOMP is due to the formation of a pore composed of the protein B-cell lymphoma (Bcl-2) protein family members Bax and Bak. In response to some apoptotic signals, Bax re-localizes from the cytosol into distinct foci on the OMM. There, Bax oligomerizes into specific structures, such as rings and arc-shaped structures, which can create large-conductance pores in the OMM [1,2,3]. Similarly, Bak (which is mostly constitutively located in the OMM) can homo-oligomerize (Figure 1).

Because of the permeabilization of the OMM, many proteins normally localized within the intermembrane space (IMS) are simultaneously released into the cytosol; these proteins are involved in the effector phase of apoptosis. In particular, (i) cytochrome *c* (CytC) mediates the organization of the apoptosome and then the activation of the caspase cascade [4]; (ii) apoptosis-inducing factor, (iii) mitochondria-associated, 1 (AIF) induces chromatin condensation; (iv) HtrA serine peptidase 2 (HTRA2) and Diablo IAP-Binding Mitochondrial Protein (SMAC/DIABLO) bind inhibitor of apoptosis (IAP), preventing the inhibition of procaspases; and endonuclease G (ENDOG) mediates DNA fragmentation [5] (Figure 1).

Investigation of the Bcl-2-mediated control of RCD revealed a role of Ca^2+^-mobilization signals. Indeed, several signals can induce RCD by the selective transfer of Ca^2+^ from the endoplasmic reticulum (ER, which acts as a store) to mitochondria [6,7,8]. When mitochondria are exposed to a pathological overload of Ca^2+^, MPT is triggered [9]. MPT is associated with the opening of the mitochondrial permeability transition pore complex (PTPC), a voltage-dependent, high-conductance channel assembled at the interface between the IMM and the OMM [10]. PTPC opening leads to the redistribution of small solutes (<1.5 kDa). The dramatic osmotic influx of water into the mitochondrial matrix during MPT collapses mitochondrial membrane potential (Δψm) and all related activities (including ATP recycling), and is followed by structural collapse (swelling). It results in the release of mitochondrial proteins from IMS which triggers the apoptotic pathway. Alternatively, the incapability of the affected cell to sustain ATP production leads to the irreversible deterioration of ion homeostasis, ultimately resulting in cell death with a necrotic morphology [11] (Figure 1).

It is believed that Ca^2+^ is the only trigger for PTPC opening, while other factors manipulate the threshold of [Ca^2+^] required for the occurrence of the event. These include sensitizers (e.g., low Δψ_m_, ROS, high matrix pH, long-chain fatty acids, atractyloside, and carboxyatractyloside) [12,13,14,15,16,17] and desensitizers (Mg^2+^, ADP, ATP, acidic matrix pH) [11]. Among the many desensitizing factors, the best characterized is cyclosporine A (CsA). This small molecule can inhibit the mitochondrial peptidyl-prolyl isomerase cyclophilin D (CypD) and to date has been considered the gold standard for evaluating the involvement of MPT in pathophysiology.

### 1.2. Current Hypotheses on PTPC Structure

The structure of the PTPC has been investigated for decades and is still not well characterized. Original studies on isolated proteins and reconstituted liposomes proposed the voltage-dependent anion channel (VDAC) on the OMM and the ADP/ATP translocase (ANT) on the IMM. The inclusion of ANT in the model is important as it possibly represents the inhibition site for ADP/ATP or the activation site for atractyloside. Notably, both proteins could be isolated in a complex containing CypD. Murine CypD is coded by the gene *Ppif*. The investigation of *Ppif*-KO mice (then ablated for CypD) robustly confirmed the involvement of CypD in PTPC composition [18]; in contrast, the genetic deletion of VDAC failed to do so. Indeed, when cells from VDAC1−/−; VDAC3−/− mice were treated with siRNA targeting VDAC2, they displayed comparable sensitivity to MPT [19].

A similar approach was conducted to investigate ANT. The knockout (KO) of the three mouse isoforms of ANT indeed showed that PTPC requires a large amount of Ca^2+^ to open, and that the inhibitory effect of ADP was lost [20]. Interestingly, in cells from ANT triple-KO animals, MPT was still sensitive to CsA, and in CypD-KO cells, MPT was still responsive to ADP [18]. The combined deletion of ANTs and CypD in a quadruple-KO model conferred resistance of MPT to [Ca^2+^] as high as 5 mM (which is considerably high), potentially indicating that the PTPC did not manifest. These experiments confirmed a role for ANT in the PTPC, but also implied that some other partners of CypD participate in the formation of the PTPC.

Recently, a new candidate was proposed as a pore-forming member of the PTPC—mitochondrial F1/FO ATP synthase (hereafter referred to as ATP synthase). Indeed, (i) genetic manipulation of ATP synthase subunits markedly affected MPT; (ii) isolated ATP synthase, or its C subunit, reconstituted in artificial bilayers generated PTPC-like currents after Ca^2+^ stimulation; (iii) ATP synthase interacts with CypD; and (iv) molecules that target ATP synthase impair MPT [21,22,23]. Furthermore, the mutagenesis of the beta subunit (β-subunit) and the oligomycin-sensitivity-conferring protein (OSCP) of ATP synthase impaired the effects of Ca^2+^ and acidic pH, respectively, on the PTPC [24,25]. ATP synthase is usually arranged in dimers that further cluster with oligomers on the IMM. PTPC-derived currents were obtained from monomers or dimers in different experimental settings, opening a debate on which portions of the complex are required [26,27]. Despite this, it was demonstrated that MPT is mediated by the rupture of ATP synthase dimers and that it was preventable by mutagenesis of the C subunit, altering the C-ring conformation [21]. CRISPR/Cas9-mediated deletion of the ATP synthase C or B subunits was still detectable, suggesting that ATP synthase might not be directly involved in the pore formation [28,29]. Later studies in cells devoid of the C subunit revealed that conductance of PTPC was significantly reduced and that the remaining current could still be inhibited by CsA or compounds targeting ANT (ADP and bongkrekic acid) [30]. Multiple hypothesis are now under evaluation to explain these apparently conflicting results: (i) ATP synthase might only indirectly regulate PTPC, by controlling crista structure and ADP levels, (ii) ATP synthase and ANT might form two independent pores among the IMM and, (iii) ANT and ATP synthase (which can interact in the so-called ATP synthasome) might be synergistically required for the proper formation of the PTPC pore.

### 1.3. Involvement of MPT in RCD Subroutines

Studies on *Ppif*-KO mice demonstrated that CypD-dependent MPT is fundamental for the activation of RCD with necrotic features. Indeed, CypD-deficient cells are resistant to necrotic cell death induced by reactive oxygen species (ROS) and Ca^2+^ overload, while stimuli that activate MOMP are insensitive to a lack of CypD [31,32]. Additionally, neuronal cell lines stably overexpressing CypD in mitochondria were prone to necrotic cell death after MPT induction, and were instead more resistant to apoptosis induced by nitric oxide (NO) or staurosporine [33]. This evidence suggests that MPT ultimately results in necrosis and does not induce other forms of cell death (Figure 1). As apoptotic cell death is an active, energy-demanding process, this conclusion is logical for all those conditions in which a marked PTPC opening is triggered, reaching the non-return point of energy depletion that engages the mechanism described above. Nevertheless, some deviation from this model should be considered. Indeed, different reports have shown that in several experimental models, different stimuli that increase intracellular ROS elicit markers of intrinsic apoptotic pathways that can be inhibited by CsA, including mitochondrial proapoptotic protein release, phosphatidylserine exposure, and DNA fragmentation [34,35,36,37,38,39,40,41,42,43,44,45,46,47,48]. This suggests that, at least under selected experimental conditions, submaximal MPT might represent an alternative path for apoptosis activation.

Furthermore, proteins that regulate MOMP are strictly connected to MPT. Several direct protein–protein interactions between Bcl-2 family members and constitutive mitochondrial proteins involved in MPT, such as ANT, VDAC, and ATP synthase, have been confirmed. Bax and Bak are also required for PTPC-dependent necrotic cell death; in fact, the loss of Bax/Bak resulted in resistance to mitochondrial calcium overload and swelling [49]. The confirmation of MPT control by these interactions was demonstrated by the evidence of Bax- and Bak-induced loss of Δψ_m_, mitochondrial swelling, and CytC release through a Ca^2+^- and CsA-dependent mechanism [50]. In addition, it was demonstrated that tBID induced transient openings of the PTPC associated with a conspicuous remodeling of mitochondrial cristae through a Bak-independent but CsA-inhibitable process [51]. ANT function is under the control of Bcl-xL expression: growth-factor-deprived cells avoid apoptosis via an efficient exchange of ADP for ATP promoted by Bcl-xL, permitting mitochondria to adapt to changes in metabolic demand [52]. Furthermore, Bcl-2 positively regulates ANT activity, while Bax inhibits it by disrupting its interaction with Bcl-2 [53].

Necroptosis is a form of regulated necrosis recently discovered under conditions in which the apoptotic pathway was inhibited, and it presents morphological features of both apoptosis and necrosis. The key upstream kinases involved in the activation of necroptosis are RIPK1 [54], RIPK3, and the substrate MLKL [55], which can be inhibited either through genetic or pharmacological methods to block this type of programmed cell death [56].

RIPK3, which is essential for TNFa-induced necrosis, can inhibit the ADP/ATP exchange mediated by ANT [57], which coincides with the loss of the CypD–ANT interaction, reduced ATP, and the induction of necrotic cell death, suggesting a role of the ANT–CypD interaction in necroptosis [58]. Bax and Bak have also been defined as mediators of necroptosis [59]; in fact, the elimination of Bax/Bak or the overexpression of Bcl-xL leads to the inhibition of the necroptotic process [60]. Furthermore, necroptosis is associated with mitochondrial CytC release and is partly sensitive to CsA inhibition [61].

The investigation of MPT over the years has revealed its importance in multiple subroutines of RCD, prompting investigation into its involvement in human pathology. In the present manuscript, we review the most recent literature discussing the role of the PTPC in human diseases caused by the dysregulation of cell death.

## 2. PTPC in Acute Conditions

### 2.1. MPT during Ischemia

Ischemia and consequent reperfusion injury (RI) are pathological manifestations in which the involvement of MPT has been robustly confirmed. Ischemia is characterized by the reduced oxygenation of a portion of a tissue (hypoxia), which results in the loss of tissue functions and eventually the activation of multiple forms of RCD. Ischemia impacts multiple organs, especially those that are more susceptible to hypoxia, such as the heart, brain, and kidney. Interestingly, not all tissues share the same susceptibility to ischemia/RI. In fact, the severity of the injury largely depends on how different types of cells, and therefore tissues, can survive under hypoxic conditions [62,63].

Mechanistically, a lack of oxygenation translates into reduced activity of the electron transport chain (ETC), and hence results in the blockage of oxidative phosphorylation (OXPHOS) and a consequent reduction in ATP recycling. In this scenario, to meet energy demands, cells upregulate anaerobic glycolysis, producing lactic acid and hydrogen ions, which results in intracellular acidosis. By neutralizing pH via the activation of Na^+^/H^+^ antiporter (NHE), the cell undergoes sodium accumulation that is counterbalanced by the reverse activity of Na^+^/Ca^2+^ exchanger (NCX). Concomitantly, the reduction in available ATP depresses the activity of Na^+^/K^+^ ATPase, plasma membrane calcium ATPase (PMCA), and sarco-/endoplasmic reticulum Ca^2+^-ATPase (SERCA), which ultimately results in an overload of cytosolic Ca^2+^ [64]. Additionally, the reduced pO_2_ causes the accumulation of electrons among different respiratory complexes, leading to the production of ROS. The concomitant increase in intracellular Ca^2+^ and ROS and the decrease in Δψ_m_ favors the induction of MPT. This mechanism has been demonstrated in cultured neonatal rat cardiomyocytes [65], hepatocytes [66], and immortalized cells [67].

Among all tissues, the brain exhibits high sensitivity to ischemia due to its glucose-dependent metabolisms [68]. Rapid and CsA-dependent mitochondrial depolarization is reported to occur early during experimental stroke in the mouse somatosensory cortex in vivo [69]. Additionally, CsA protects the retinal ganglion from cell death during acute intraocular pressure (IOP) elevation, a peculiar inducer of ischemia [70]. Furthermore, CsA administration attenuates hypoxic–ischemic brain injury in newborn rats induced by unilateral carotid artery ligation [71].

Despite this, MPT is commonly believed to occur to a minimal extent during ischemia. Indeed, as described, hypoxia leads to the accumulation of protons and ADP, which are strong inhibitors of PTPC. During ischemia, the manifestation of MPT depends on the subtle equilibrium between PTPC inducers (Ca^2+^ and ROS) and inhibitors (protons and ADP), which accumulate as a result of impaired mitochondrial respiration and excess glycolysis (Figure 2). Furthermore, the adaptive response to hypoxia-mediated by HIF1a affects PTPC opening by regulating hexokinase II (HKII) levels [72], another reported PTPC regulator. Indeed, when HIF1a is stabilized (e.g., following hypoxia or GSK360A administration), HKII protein expression significantly increases in the mitochondrial fraction, and this is directly involved in the cytoprotective effect started by HIF1a stabilization. Genetic depletion of HKII completely abolished this path even in the presence of an activated HIF1a [72]. Nevertheless, HKII is considered an inhibitor of the PTPC only when bound to the OMM (especially to VDAC), rather than when generally overexpressed in mitochondria.

Nevertheless, under this condition, Ca^2+^ overload may induce MPT-independent cell death as a result of the excessive activation of Ca^2+^-dependent enzymes such as phospholipases, proteases, and endonucleases [73,74]. Noticeably, during myocardial infarction, the activation of calpains, Ca^2+^-dependent cysteine proteases, results in myofibril disruption, thus promoting hypercontracture in the heart, which consists of sustained shortening and stiffening of the myocardium [75,76].

### 2.2. Role of MPT in Reperfusion Injury

The situation dramatically changes during tissue reperfusion. Indeed, the restoration of pO_2_ recovers respiration, ATP synthesis, and the activity of plasma membrane pumps, which re-equilibrate intracellular pH. As a result, the inhibition of MPT by ADP and protons is removed, lowering the threshold for PTPC opening. In addition, reperfusion stimulates ROS production by multiple sources [77]. At the mitochondrial level, succinate is observed to accumulate in the mitochondrial matrix during ischemia and it was proposed to stimulate ROS production from complex I through reverse electron transport (RET), at the time of reperfusion [78]. Other pieces of evidence suggest that ETC conditions are not favorable for RET at the beginning of reperfusion, and that a burst of ROS production only occurs after a first wave of MPT. Besides mitochondria, reperfusion induces ROS via other enzymes including (but not limited to) xanthine oxidase, NADPH oxidase, and nitric oxide synthase [77]. These are believed to further stimulate ROS generation in the mitochondria, forming a vicious cycle and making mitochondria de facto the largest ROS source during reperfusion [79,80]. The elevated superoxide anion (O_2_^−^) reacts with nitric oxide (NO), producing the highly reactive peroxynitrite. This leads to a reduction in the availability of NO (which is a potent vasodilator) and causes the accumulation of neutrophils [77]. Furthermore, oxidative damage causes lipid peroxidation, DNA damage, and enzyme denaturation and activates the innate anti-inflammatory response, aggravating reperfusion injury (RI) [81].

The occurrence of MPT during the reperfusion phase was demonstrated by the experiments of Griffith and Halestrap in 1995. They showed that mitochondrial accumulation of radioactive deoxy glucose (hot-dog), which can pass through the IMM only during MPT, did not occur in isolated hearts undergoing ischemia, but was significantly induced (and inhibitable by CsA) during reperfusion [82].

In the past decade, several in vitro and in vivo studies have confirmed the involvement of MPT in ischemia/RI [83]. The administration of CsA (as well as its analog FK506) has been shown to protect against ischemia/RI in multiple animal models and in multiple tissues, including cardiac and skeletal muscle, brain, kidney, liver, lungs, and testis [84,85,86,87,88,89]. *Ppif*-KO mice are significantly protected from ischemia/RI in both cardiac muscle and the brain, in terms of both tissue function and survival rates [31,32,90]. In addition to CypD manipulation, genetic interference in mechanisms of Ca^2+^ homeostasis proved the importance of MPT in ischemia/RI. Cardiac-specific ablation of NCX significantly decreased ischemia/RI in isolated hearts [91]. Similarly, mice overexpressing *Bcl2* have reduced [Ca^2+^]_m_ accumulation and are therefore protected from myocardial ischemia/RI [92]. Accordingly, tissue-specific overexpression of the mitochondrial Na^+^/Ca^2+^ exchanger (TER-NCLX) in cardiac muscle accentuates the extrusion of Ca^2+^ from mitochondria to the cytosol and then suppresses [Ca^2+^]_m_, decreasing the sensitivity of cardiomyocytes to PTPC opening. Hearts from TER-NCLX mice also display protection from left coronary artery ligation-induced ischemia/RI [93]. Additionally, impairing mitochondrial calcium uniporter (MCU), using the inhibitor Ru360 [94] or in MCU-KO mice [95], lowers the uptake of mitochondrial Ca^2+^ and is correlated with increased brain and heart function. We recently demonstrated that inhibiting ATP synthase via N,N-dicyclohexylcarbodiimide (DCCD) partially recovered contractility of isolated heart exposure to ischemia/RI by Langhendorff apparatus. Most interestingly, we observed that serum levels of the C subunit in patients with ST-segment elevation myocardial infarction (STEMI) correlated to several surrogate markers of myocardial reperfusion [96].

### 2.3. Role of MPT in Acute Kidney Injury

Renal tissue is also known to be affected by MPT in multiple conditions. Acute kidney failure (mostly known as acute kidney injury, AKI) is often characterized by extensive necrosis. The protective effect of CsA or CypD inactivation on experimental ischemia/RI has been largely reported in kidneys, as previously stated. Atherosclerotic renal artery stenosis (ARAS) is probably the most frequent cause of ischemia/RI-related AKI. The outcomes of experimental ARAS are significantly improved by exposure to the mitochondrial-targeted peptide Elamipretide. This peptide displays multiple protective functions, including the buffering of ROS and the stabilization of the structural mitochondrial lipid cardiolipin. Specifically, in experimental ARAS, the protective effect of Elamipretide is believed to be mediated by desensitization to PTPC opening [97].

Renal tissue can also undergo, at nominal pO_2_, conditions resembling manifestation of RI and that appear to be dependent on MPT. For example, an important cause of AKI with extended necrosis is exposure to drugs (e.g., FANS or chemotherapy) and crystal nephropathies. The etiology of these ischemic-like conditions is not yet fully comprehended, although it is proposed to act through excess ROS production, which ultimately triggers PTPC opening. In support of this model, under conditions of oxidative stress, glycogen synthase kinase-3β (GSK3β), an interactor and regulator of the putative PTPC, translocates from the cytosol to mitochondria in a VDAC2-dependent manner, promoting PTPC opening [98]. Inhibition of GSK3β promotes resistance to MPT in mice undergoing paraquat- or diclofenac-induced nephrotoxicity [99,100]. Interestingly, it is now well recognized that cell death during AKI is significantly dependent on necroptosis, a finding largely confirmed by investigations on Mlkl-KO and Ripk3-KO animals [101]. Cisplatin-induced AKI is protected by the inactivation of both Ripk3 and CypD. *Ripk3/Ppif*-double-KO models are more protected than the models with either KO [102], suggesting that MPT and Ripk3 contribute to RCD by independent but concomitant pathways. A recent study by Mulay et al. using CypD-KO and Mlkl-KO mice showed that experimental conditions mimicking acute oxalosis (crystal nephropathy characterized by sudden increases in serum oxalate levels) resulted in kidney failure with necroptotic features that was strongly dependent on MPT [103]. That study, however, did not report significant differences among the *Ppif* -KO, *Mlkl*-KO and *Ppif*/*Mlkl*-double-KO mice in terms of protection from cell death.

## 3. PTPC in Degenerative Conditions

Degenerative disorders are human diseases characterized by the chronic loss of fully differentiated cells, which leads to the progressive impairment of the structure and/or function of the affected tissue/organ. Among the most frequent and probably most investigated degenerative disorders are conditions that affect the CNS or the skeletal or cardiac muscle. As previously mentioned, these organs all require large amounts of energy; therefore, it is not surprising that the mutations most frequently associated with degenerative disorders are linked to mitochondria. Intriguingly, many of these conditions share impaired mitochondrial respiration and ATP production, increased ROS production, and altered PTPC sensitivity. The involvement of MPT in these disorders appears to be more complex than the involvement of MPT in ischemia/RI in terms of molecular and cellular interactions. We therefore summarize the major observations made in different degenerative conditions.

### 3.1. Role of the PTPC in Protein-Aggregation-Related Neurodegenerative Diseases (NDs)

The primary features of neurodegeneration are the abnormal presence and accumulation of mutant and/or damaged proteins. Protein aggregation is the main cause of changes in the intracellular environment such as oxidative status, impaired protein quality control system, transcriptional alteration, and mitochondrial dysfunction. All these variations critically contribute to the pathogenesis of NDs and culminate in neuronal cell death. Among the diverse types of proteins that aggregate, amyloid-beta (Aβ), Tau, and alpha-synuclein (αSyn) are the most commonly studied and represent the primary cause of sporadic and familial Alzheimer’s disease (AD) and Parkinson’s disease (PD), the most prevalent NDs.

Interestingly, Aβ, Tau protein, and αSyn are found in the mitochondria of patients affected by AD or PD [104,105,106] or their related animal models [107,108,109]. In particular, these protein aggregates have been observed to colocalize or directly interact with multiple partners of the PTPC, such as CypD, VDAC, ANT, and ATP synthase [104,107,109,110].

Additionally, Aβ exposure in cultured cortical neural progenitor cells induces PTPC opening [111]. In particular, short Aβ exposure led to decreased cell proliferation. However, when the exposure and thus PTPC opening were prolonged, CsA-inhibitable necrotic RCD was activated. Consistent with this result, intravital multiphoton imaging of AD mouse models demonstrated that near senile Aβ plaques, mitochondria showed severe structural and functional abnormalities, suggesting that senile plaques are the main source of toxicity in vivo [112]. Aβ plaques and phosphorylated Tau interact with VDAC, leading to mitochondrial dysfunction [105]. Similarly, a specific Tau fragment (NH2-26-44 fragment) affects OXPHOS and mitochondrial dynamics by interacting with ANT and impairing PTPC regulation [113].

Furthermore, αSyn oligomers move into mitochondria and colocalize with ATP synthase, inducing its oxidation concomitantly with increased PTPC opening, mitochondrial swelling, and necrosis activation [114]. Additionally, a PD mouse model characterized as having a mutant human αSyn (Thy1-hαSyn-A53T tg mice) proved that αSyn associates with neuronal mitochondria and interacts with VDAC and CypD in vivo [115]. This work directly linked motor abnormalities and neuropathology to the PTPC.

The study of rare inherited mutations in PD has provided insight into the molecular mechanisms of mitophagy, the regulated delivery of dysfunctional mitochondria to lysosomes via autophagic machinery. Multiple PD-related genes have been identified, among which the mitochondrial kinase PINK1 and the cytosolic E3 ubiquitin ligase Parkin are the most characterized. The PTPC may be involved in mitophagy-mediated quality control processes, which play a critical role in conserving neuronal health and function. Indeed, it has been suggested that the opening of the PTPC regulates mitochondrial depolarization and subsequent mitochondrial degradation in autophagosomes. Furthermore, CsA and its analogs block autophagosome formation, and the alteration of PTPC opening has been unveiled in models lacking PINK1, Parkin, and DJ1, which are components of the best-characterized stress-induced mitophagy pathways [116,117]. Consistently, the downregulation of PINK1 in mouse neurons resulted in altered mitochondrial morphology and function, ROS production, and finally PTPC opening. All these events were accompanied by the induction of mitochondrial autophagy [118].

Similarly, PINK1-deficient neurons showed selective increases in mitochondrial Ca^2+^, PTPC opening, and defective mitochondrial respiration. In addition, the inhibition of PTPC opening was found to be sufficient to rescue the mitochondrial impairments observed in *Pink1**−*/− cells [119]. ROS production and PTPC opening were increased in primary mouse embryonic fibroblasts (MEFs) and brains from *Park7*−/− (the gene coding for DJ-1) mice compared with wild-type (WT) samples. In contrast, antioxidant molecules decreased ROS levels and PTPC opening. Interestingly, in contrast to *Pink1*−/− cells, the lack of DJ-1 did not affect mitochondrial respiration and Ca^2+^ dynamics, suggesting that DJ-1 has a possible antioxidant role. Finally, Parkinsonian toxins (such as 6-hydroxydopamine and neurotoxin1-methyl-4-phenyl-1,2,3,6-tetrahydropyridine) are widely employed as in vivo and in vitro chemical models of PD and have been found to be potent activators of both PTPC and mitophagic processes [120,121]. Notably, both VDAC and ANT were demonstrated to be required for proper mitophagy [122,123]. Finally, recent pieces of evidence demonstrated that human samples obtained by AD-affected patients displayed inhibited damaged mitochondrial clearance [124] and that mitophagy activation diminished insoluble Aβ and Tau hyperphosphorylation to revert cognitive impairments in an AD mouse model [125,126] (Figure 2).

Taken together, these findings suggest that the mitochondrial accumulation of disease-specific protein aggregates might favor MPT via direct interactions or mitophagy impairment, which leads to the accumulation of mitochondria prone to PTPC opening and RCD (Figure 2).

### 3.2. Amyotrophic Lateral Sclerosis (ALS) and PTPC

ALS is the most common neuromuscular degenerative disease affecting adults. While several works have suggested a protein-aggregation origin, this progressive and severely disabling fatal neurological disease is generally considered to have multifactorial causes. Currently, there are no cures or effective treatments for ALS, and the molecular pathogenesis of ALS is poorly understood.

Recent findings show that mitochondrial perturbations are implicated in the pathogenesis and progression of ALS. Altered fission–fusion dynamics, altered mitochondrial Ca^2+^ homeostasis, excessive oxidative stress, reduced OXPHOS activity, and decreased proapoptotic factor release have been found in ALS models in vitro and in vivo. Additionally, the PTPC is emerging as a critical player in ALS. In a transgenic model of ALS (G93Ahigh), a profound alteration in mitochondrial structures with increased PTPC activity was observed [127]. Interestingly, dendritic mitochondria from the same ALS animal model displayed increased contact sites between the IMM and OMM. This conformation might favor the formation of the PTPC. Furthermore, the deletion of CypD delayed disease onset and extended the survival of transgenic ALS mice [127]. Consistent with this finding, the exposure of two independent ALS murine models to the novel PTPC inhibitor GNX-4728 protected against motor neuron degeneration and mitochondrial impairment and promoted their survival nearly 2-fold [128].

Preclinical studies have shown that olesoxime, a member of the cholesteroloxime family, improves the survival of neural cells and reduces the effects of oxidative stress by modulating the PTPC. In particular, this compound concentrates in mitochondrial compartments, where it binds the PTPC interactors VDAC and mitochondrial translocator protein (TSPO). Following olesoxime binding, the PTPC was desensitized, leading to neural cell protection both in vitro and in vivo [129,130]. Accordingly, the potent antioxidant and inhibitor of PTPC edaravone (Radicut™) has been approved for the treatment of ALS [131]. Compounds derived from cinnamic anilides such as GNX-4728 and GNX-4975 also represent an MPT-based treatment of ALS. In a murine model of ALS, these compounds delayed the onset of symptoms, increased lifespan, and reduced the inflammatory response [128].

### 3.3. Multiple Sclerosis (MS) and PTPC

MS is the most common primary demyelinating disease of the brain. MS is an inflammatory T-cell-mediated autoimmune disease characterized by progressive demyelination, gliosis (scarring), and neuronal loss [132]. Recently, mitochondrial dysfunction has been increasingly linked to the pathogenesis of MS. Additionally, impaired mitochondrial enzyme complex activity [133,134], increased oxidative stress [135], altered mitochondrial DNA [136], impaired quality control systems [137,138], and abnormal mitochondrial number and morphology have been described in MS patients [133] and in vivo MS mouse models [139,140]. In this context, it was discovered that the PTPC might also contribute to MS; indeed, CypD-KO mice with experimental autoimmune encephalomyelitis (EAE), a commonly used animal model for MS, recovered from their induced disabilities. Furthermore, axonal damage was decreased, and the mitochondria of cultured CypD-KO neurons accumulated higher levels of Ca^2+^ and were more resistant to oxidative stress compared to WT [141]. Subsequent studies confirmed this finding and demonstrated that the selective inhibition of PTPC exerted neuroprotective effects on the EAE model by increasing mitochondrial function, reducing oxidative stress, and blocking mitochondrial swelling and Ca^2+^-mediated PTPC formation [142]. The p66Shc protein may also modulate the mitochondrial dynamics and PTPC opening that occur during neurodegeneration in EAE. P66Shc is a product of the ShcA gene normally localized in the cytoplasm. However, once phosphorylated by protein kinase C-Beta and following interaction with PIN1, P66Shc moves to the mitochondria, where it regulates distinct cellular processes such as apoptosis and autophagy [143,144]. Furthermore, in the mitochondrial compartment, p66Shc works as a ROS amplifier by generating mitochondrial ROS to induce PTP opening. Consistent with this function, when EAE was induced in p66Shc-KO (*p66Shc−/−*) mice, the clinical symptoms (manifested as limb weakness and paralysis) of these model mice were less severe than those of WT mice [145]. The fact that the onset and development of EAE in p66Shc/Cyc-D-double-KO mice were identical to those observed in *p66Shc−/−* mice validates the role of the p66Shc-PTPC pathway in neurodegeneration and confirms that once activated, p66Shc interacts with PTPC to promote its opening [146].

### 3.4. PTPC in Muscular Dystrophies (MDs)

MDs refer to a clinically and genetically heterogeneous group of degenerative muscle diseases, manifested primarily as the progressive weakness and degeneration of skeletal muscles that control movement, resulting in severe pain, disability, and ultimately death. Some forms of MD also affect cardiac muscle [147,148]. Current therapies to treat muscle degenerative diseases are still limited by poor targeting, although promising new therapeutic directions remain [149].

The most common and severe form of human muscular dystrophy is Duchenne MD (DMD), which is an X-linked recessive genetic disorder associated with respiratory complications and cardiac dysfunction [150]. The disease is caused by a genetic defect—the absence of the cytoskeletal protein dystrophin, the primary function of which is to link the myofiber cytoskeleton to the extracellular matrix, stabilizing the sarcolemma [151,152,153]. Although the gene underlying the disorder was identified in 1987 [154], the pathophysiology leading to disease remains unclear. Numerous mitochondrial alterations are correlated with dystrophic conditions, including impaired ATP production, substrate handling, Ca^2+^ buffering capacity, and elevated ROS production. The mitochondria in muscular fibers from dystrophic mdx mice (a murine model of DMD) displayed a significantly shorter time to MPT induction in response to Ca^2+^ than WT mice [155]. In *C. elegans* and zebrafish models of DMD, mitochondrial fragmentation is detectable before overt signs of muscular degeneration, and CsA feeding delays muscle degeneration [156]. Cyclosporine also inhibits calcineurin, a signaling protein involved in skeletal muscle, and its inhibition was found to worsen muscular dystrophy in an mdx mouse model [157,158]. Nonetheless, the deletion of *Ppif* and the administration of Debio-025 (a CsA inhibitor with no effect on calcineurin) prevented dystrophic conditions in mdx and d-sarcoglycan-KO *(Scgd−/−)* animals [159,160]. Additionally, several lines of evidence have shown that Ppif deletion is protective in *Col6a1−/−* mice [161].

Mutations of collagen VI (ColVI) genes encoding the extracellular matrix protein, which is abundant in skeletal muscle, cause three muscle diseases in humans: Ullrich congenital muscular dystrophy (UCMD), Bethlem myopathy (BM), and the recently identified myosclerosis myopathy (MM) [162,163]. These collagen VI myopathies are inherited muscle diseases that share mitochondrial dysfunction due to altered PTPC opening [164]. Mouse models lacking collagen VI (*Col6a1−/−*) display an early onset myopathic phenotype correlated with ultrastructure defects in mitochondria and the sarcoplasmic reticulum (SR), altered mitochondria caused by eventual inappropriate PTPC opening, and elevated muscle fiber apoptosis [161,165,166]. In addition, the absence of ColVI led to a marked decrease in the expression of proapoptotic Bcl-2, which may synergize with calcium to enhance the opening of the PTPC and eventually promote the release of mitochondrial proapoptotic factors [167].

In light of this fact, the altered expression of collagen in *Col6a1−/−* mice and in BM and UCMD patients correlates with enhanced PTPC opening, resulting in the functional and ultrastructural deficiency of mitochondria, followed by impaired autophagy, [168]. As such, autophagy is altered in MDs, and autophagy activation due to low amino acid intake improved the skeletal muscle phenotype of a DMD mouse model (mdx), [169]. Consistently, pharmacological treatment with CsA has been reported to dramatically recover myofiber degeneration in a Col6a1−/− mouse model and UCMD patients [161,166]. Additionally, it has been demonstrated that mitochondria-mediated cell death can be reduced by CsA in Ullrich congenital muscular dystrophy models [167].

## 4. Mitochondrial Disorders

Mitochondrial diseases are a clinically heterogeneous group of disorders that arise because of mitochondrial respiratory chain dysfunction. These diseases are caused by mutations in nuclear DNA (nDNA) and in mitochondrial DNA (mtDNA). Although mitochondrial diseases can involve any organ or tissue, they characteristically involve multiple systems, typically affect organs that are highly dependent on aerobic metabolism and are often relentlessly progressive with high morbidity and mortality. Like other degenerative diseases, the involvement of MPT in mitochondrial diseases has been proposed.

Mutations in mtDNA are responsible for the etiology of the most frequent mitochondrial diseases, especially Leber’s hereditary optic neuropathy (LHON); neurogenic muscle weakness, ataxia, and retinitis pigmentosa (NARP); mitochondrial encephalomyopathy, lactic acidosis, and stroke-like episodes (MELAS); and myoclonic epilepsy with ragged red fibers (MERRF). These alterations most frequently involve single-base mutations in genes encoding components of respiratory complex I, ATP synthase, transfer RNA, and DNA polymerase, but also may be caused by large deletions of mtDNA. Hybrids carrying mtDNA mutations associated with LHON, NARP, MELAS, and MERRF displayed poor resistance to oxidative stress, which could be prevented by the administration of CsA or the deprivation of extracellular Ca^2+^ [170]. Accordingly, many of these mutations lower the threshold for PTPC opening in response to Ca^2+^ and ROS [171,172].

Mutations in nDNA associated with mitochondrial diseases have also been also related to alterations in MPT activity. Mutations in *leucine-rich pentatricopeptide repeat containing (LRPPRC)*, a protein involved in the maturation and stability of mitochondrial RNA [173], cause the French-Canadian variant of Leigh syndrome. Loss of LRPPRC results in defects in the assembly of respiratory complexes IV and V, leading to severe metabolic alterations. Fibroblasts isolated from patients presenting LRPPRC inactivation displayed multiple types of mitochondrial dysfunction, including a reduced threshold for Ca^2+^-induced MPT [174]. Interestingly, all the mitochondrial alterations mentioned so far are often associated with impaired or unstable assembly of ATP synthase, providing significant clues to its involvement in MPT. Direct investigations of ATP synthase alterations and MPT in the etiology of mitochondrial disease have not been performed. Other significant nuclear genes relating mitochondrial diseases to MPT are optic atrophy 1 (OPA1) and Spastic paraplegia 7 (SPG7). OPA1 is an essential protein involved in the fusion and cristae arrangement of the IMM. Its mutations manifest clinically as optic atrophy, ataxia, and deafness [175]. SPG7 (paraplegin) is an ATP-dependent zinc metalloprotease located in the mitochondrial matrix, and its mutations are associated with chronic progressive ophthalmoplegia [176]. Interestingly, both proteins positively regulate the MPT threshold in response to Ca^2+^ induction, and their inactivation significantly inhibits PTPC opening [177,178]. Mechanisms by which OPA1 and SPG7 act on PTPC are still to be defined. While the role of OPA1 seems to be dependent on its control of crista morphology [179], SPG7 appears to regulate the amount of Ca^2+^ available for MPT via regulation of mitochondrial calcium uniporter assembly [180]. Taken together, this evidence indicates that PTPC might be a significant target for mitochondrial disease; however, the mechanism by which MPT influences these diseases is still poorly understood and might differ among syndromes, calling for further investigations in more complex experimental models.

## 5. PTPC in Nonalcoholic Fatty Liver Disease

Nonalcoholic fatty liver disease (NAFLD) is among the most prevalent chronic liver diseases in both children and adults, and is predicted to be the primary cause for liver transplants by 2020 [181]. NAFLD is characterized by an accumulation of fat (steatosis) in the liver, which can progress to inflammatory NASH and into more severe stages: fibrosis, cirrhosis, and hepatocellular carcinoma [182]. The prevalence of NAFLD has risen rapidly in Western societies, particularly in most of the European countries, due to an increase in mass consumption of highly processed ready-made food, which is rich in fructose and saturated fat. NAFLD has risen rapidly in parallel with the recent surge in metabolic-related diseases such as obesity and type 2 diabetes mellitus (T2DM), which have been indicated as risk factors for the prevalence and progression of NAFLD. Despite the attempts of the liver to recover from fat accumulation, in the long run, mitochondrial adaptation is insufficient to prevent lipotoxicity due to continuous FFA accumulation [183]. At this later time point, mitochondria present alterations in the OXPHOS complexes, mitochondrial membrane potential, reduced ATP synthesis, and induced PTPC opening [183]. Opening of the PTPC may be the basis of the steatosis-induced apoptosis of hepatocytes observed in vitro, and may be related to the steatosis in NAFLD of human beings [184]. It has been postulated that stimulated PTPC opening in a rat model of NAFLD is the result of increased Bax expression and aberrant Bcl-2/Bax ratio. This seems to be an important mechanism of the mitochondrial damage in hepatocytes that occurs in NAFLD [185]. Wang et al. proposed that the overexpression of mitochondrial hepatic CypD induced mitochondrial stress and could be an early event that leads to the liver steatosis [186]. They found that overexpression of CypD is manifested in mitochondrial swelling and increased mitochondrial ROS production. Such mitochondrial perturbations provoke ER stress through Ca^2+^/p38 MAPK activation, finally resulting in the increase of SREBP1c-mediated synthesis of triglycerides. Interestingly, in mice fed with a high-fat diet, the increased level of CypD was observed earlier than triglyceride accumulation in the liver. Moreover, Wang et al. speculated that CypD knockout or pharmacological inhibition of CypD could ameliorate triglyceride accumulation in HFD-fed mice [186]. On the other hand, the observations of Lazarin et al. indicated that mitochondria isolated from livers of monosodium l-glutamate obese rats were less susceptible to the opening of PTPC by calcium [187]. Regardless of several pieces of evidence for the involvement of the MPT in the NAFLD animal models, there is no direct evidence supporting the role of PTPC and especially CypD in NAFLD in humans.

## 6. PTPC in Cancer

One of the earliest established hallmarks of cancers is the resistance of transformed cells to RCD. Considering the discussed role of MPT in RCD, it is straightforward to hypothesize that the alteration of the PTPC machinery is involved in the establishment of neoplasia. According to this hypothesis, MPT is predicted to have a tumor-controlling mechanism, and its suppression is required for tumor development. It is harder to prove that a phenomenon does not occur than it is to prove that it does occur. Indeed, strong direct evidence confirming or confuting this hypothesis is lacking. Experiments based on the genetic manipulation of CypD for the other disease types discussed should provide significant evidence, but nothing of this kind has yet to be reported. Still, reduced MPT in transformed cells could be predicted by a large number of findings, as discussed below.

Another significant hallmark of cancer is metabolic rewiring, especially the abnormal increase in the glycolytic rate at almost normal pO_2_ (Warburg effect). This large glucose consumption causes the significant conversion of pyruvate into lactate, resulting in intracellular acidification and leading to PTPC desensitization similar to that observed in ischemia [6]. Furthermore, many solid tumors develop a hypoxic area that, analogous to ischemic tissue, also promotes Hif1a accumulation and HKII-mediated desensitization of PTPC. The Warburg effect then allows the hypoxic tumor to limit Pi and Ca^2+^ accumulation, similarly to ischemia, favoring the inhibitory effect of low pH. If, by analogy to ischemia, an increase in [Ca^2+^] could be expected, it is true that transformed cells have lower intracellular Ca^2+^ (Figure 2). Indeed, it has been shown that H-RAS-driven transformation is concomitant with a progressive reduction in the amount of intracellular Ca^2+^ [188]. Bcl-xL has been shown to negatively regulate PTPC opening by directly interacting with VDAC [189]. Additionally, Bcl-xL can interact with the β subunit of ATP synthase to promote its synthase activity and inhibit PTPC [190].

Furthermore, many other oncogenes and oncosuppressor genes regulate Ca^2+^. Among the most characterized is Bcl-2, which can impair [Ca^2+^] in the ER lumen and its transfer to mitochondria via multiple proposed mechanisms [191]. Additionally, many tumors show alterations in the PI3K pathway. Members of this pathway, AKT and PTEN, can localize at mitochondrion-ER contact sites, especially with Ip3R, to alter its activity [192,193,194,195,196]. This effect is coordinated by the oncosuppressor PML [197,198,199]. Loss of PML or PTEN and activation of AKT (conditions prototypical of multiple tumor types) lead to the inactivation of IP3R, which also limits the Ca^2+^ available to mitochondria. Additionally, AKT phosphorylates GSK3β (altered in several cancer types), resulting in its inactivation and, as discussed, PTPC desensitization [200]. Finally, the master oncosuppressor p53 interacts with SERCA to maintain a high level of ER [Ca^2+^]. Loss or inactivation of p53 (one of the most common alterations in cancer) impairs [Ca^2+^] and results in reduced sensitivity to MPT-mediated RCD [201,202,203,204,205]. In addition, p53 can localize to mitochondria and interact with CypD to favor MPT and necrosis.

Interestingly, a network of chaperones seems to interact with CypD (which could be considered a chaperone itself) to modulate PTPC. In particular, HSP90, HSP60, and DnaJC15, which are often overexpressed in tumors, have been shown to interact with CypD, leading to their inhibition (then mimicking CsA) and suppressing RCD initiation [206,207,208]. These findings are supportive of the hypothesis, but contradict other strong evidence. First, transformed cells often have increased levels of ROS, not of a significant increase in multiple scavenging systems. It is currently accepted that elevated ROS can act as a mitogenic signal in tumors to support proliferation [209,210]. In addition, studies in cancer cell lines and tumor models have shown that different PTPC members are overexpressed, especially TSPO, VDAC, and ANT, possibly to favor their specific metabolic condition [16,211,212,213,214,215,216,217]. We can, therefore, speculate that tumor cells can survive pressure selection by the highly regulated suppression of MPT, which allows the maintenance of MPT-related features with potential mitogenic effects (Figure 2).

## 7. Potential of PTPC Targeting and Concluding Remarks

Because of the presented evidence (and much more), MPT has been investigated for the treatment of human disease in multiple clinical trials. CsA entered a trial procedure that lasted 15 years and ended with failure at phase III in cardiac RI. Indeed, the CIRCUS [218] and CYCLE [42] trials consisting of a single intravenous bolus of CsA (2.5 mg/kg) before revascularization had no effect on ST-segment resolution or cardiac enzymes and did not improve clinical outcome. These findings were reconsidered by a study by Piot et al., who saw hope in the use of CsA to treat RI. Similarly, TSPO targeting by 3,5-Seco-4-nor-cholestan-5-one oxime-3-ol (TRO40303) was tested in clinical trials, due to promising cardioprotective effects on a rat model of cardiac ischemia. However, the desensitization of PTPC opening seemed to be secondary to its remarkable antioxidant properties [219]. Indeed, studies on TSPO-KO mice showed that the protein was dispensable in models of ischemia/RI [220]. However, the safety and efficacy of this drug were evaluated a few years later in patients undergoing percutaneous coronary intervention (PCI). This multicenter, double-blinded, phase II study (MITOCARE) showed the inefficacy of the compound in reducing or limiting RI [221]. Additionally, an oral version of CsA was tested for the treatment of LHON but failed to reach the primary endpoint of the study, although it delayed the onset of the disease. [222]. There are multiple reasons that CsA-based trials have failed to reproduce the protection reported in preclinical studies, but a discussion of these failures is not the purpose of this review. The investigation of novel compounds able to target the PTPC is ongoing, and many investigations have already yielded promising results in preclinical studies. Most of these are designed to be more potent and specific inhibitors of CypD, including the small molecules C-9, C-19, and C-31, which have already been proven to be protective in models of AD, acute pancreatitis, and hepatic injury [223,224,225].

Library screening has also identified ML-404 [226] and N-phenylbenzamide [227] as CypD-independent, offering the possibility to combine their use with CsA to limit its side effects. It is of interest that ATP synthase is now the subject of investigation in these terms. Oligomycin and DCCD, which target the C subunit, displayed powerful MPT inhibition in vitro, but they also depleted mitochondrial ATP, causing an additional injury [21].

We recently generated a library of small molecules that target the C subunit by modifying the functional core of oligomycin and obtained new patented compounds able to notably reduce reperfusion damage in animal models of global ischemia without interfering with ATP production [22]. Finally, the natural hormone melatonin is of great interest. This hormone can act as an antioxidant and can modulate the PTPC, although its exact mechanism is currently under investigation [228,229]. Currently, melatonin is considered the safest drug that can be used as a PTPC inhibitor, and we will probably see an increasing number of investigations on this molecule in the future.

## Figures and Tables

**Figure 1 biomolecules-10-00998-f001:**
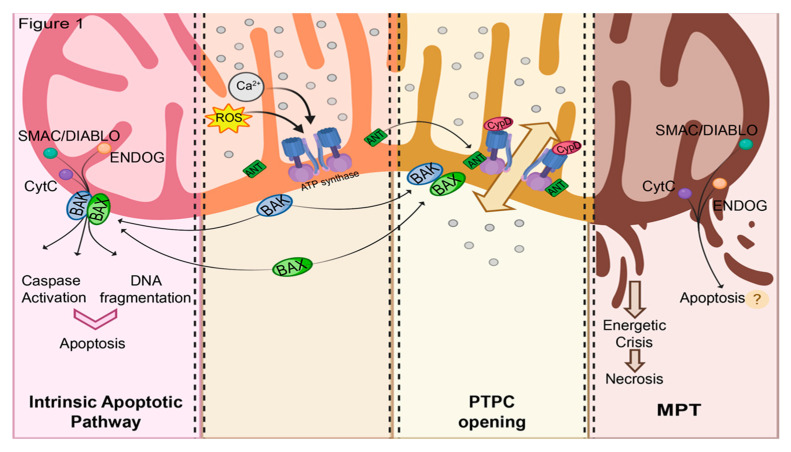
Major molecular paths in mitochondria-related regulated cell death (RCD). Mitochondrial calcium overload and ROS levels can trigger either the activation of intrinsic apoptotic pathway (left side) through the recruitment of Bcl-2 family proteins at the mitochondria, or permeability transition pore complex (PTPC) formation which could lead to mitochondrial outer membrane permeabilization (MOMP), energetic imbalance, and subsequent release of proapoptotic cofactors from the inter membrane space, such as SMAC/DIABLO, CytC, and ENDOG (right side).

**Figure 2 biomolecules-10-00998-f002:**
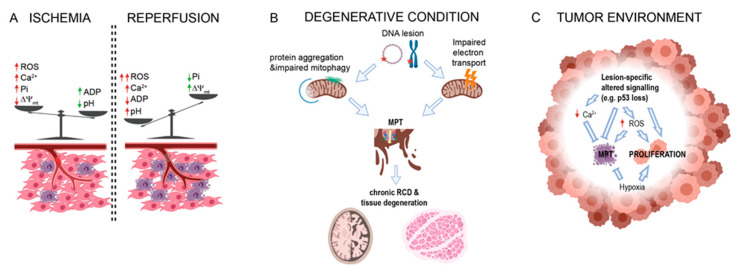
MPT alterations in human diseases. (**A**) Effect of ischemia and reperfusion in levels of MPT-regulating factor in insurgence of RCD (purple cells). (**B**) Schematic representation of the effect of mutations in mitochondrial or nuclear DNA (represented by circular DNA or chromosome, respectively) in human diseases characterized by degeneration of neuronal or muscular tissue. (**C**) Representation of major alterations in MPT regulators detected in tumor microenvironment.

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
