# Peer review of "Physiopathology of the Permeability Transition Pore: Molecular Mechanisms in Human Pathology"

_biomolecules, 2020, doi:10.3390/biom10070998_

Round 1

Reviewer 1 Report

This review article describes MPT and its involvement in etiology of human diseases. The topic is appropriate as the pathological significance of mitochondrial dysfunction and MPT is well recognized. Some changes are needed to improve this review.

Comments:

1. Although it is understood that the main body of this review manuscript is the description of pathological involvements of PTPC, a lot of information is cramped in the Introduction and therefore it got quite long. Dividing into additional headings or subheadings based on content will provide better guidance to the readers.

2. There are too many very short paragraphs (1 or 2 sentences). Sometimes, those short paragraphs seem to be used as a sort of subheadings. If so, just put them as actual subheadings. Otherwise, incorporate them into the paragraphs above or below. For example, there should be no new paragraph for lines 109-110, and lines 111-113 can be incorporated into the paragraph below. Content-wise, the paragraph (line 130-132) doesn’t seem to belong anywhere. Either expand it or move to appropriate place.  

3. There are many missing references. Some examples are: lines 130-132, lines 175, 302, 304, 320…. and Concluding Remarks.

4. Line 90-102 describes ATP synthase as PTP. However, studies from Walker group showed that MPT occurs without c subunit or peripheral stalk, suggesting that ATP synthase is not (a part of) the PTP (PNAS 2017, 114: 3404 and 9086). This information should be included.

5. Line 170, Na/K-ATPase, how about PMCA (plasma membrane Ca-ATPase)?

6. Line 198, different sources [47], indicating the ROS sources other than mitochondria. Then, need to specify the sources.   Also, a large volume of literature has shown that reperfusion increases ROS from mitochondria. More recently, Michael Murphy's group showed that the succinate accumulated during ischemia produces superoxide by reverse electron transport, although Halestrep's group refutes it. Regardless, mitochondria have been shown to be the major source of ROS during reperfusion. This information needs to be included.

7. Line 32, change “under” to “below”.

8. Line 34, remove “the” from “the that”

9. Line 52, the “c” in the cytochrome c should be lower case italic

10. Line 73. Chang PTOC to PTPC

11. Line 113, “the mechanism previously described”. Change to “the aforementioned mechanism” or “the mechanism described above”. If it really meant “previously”, which study? Need reference.

12. Line 124, “to OMM swelling and to mitochondrial Ca overload”. OMM does not swell. Change to “to mitochondrial Ca overload and swelling”

13. Line 152, change RDC to RCD.

14. Line 187, should explain HKII. Hexokinase?

15. Line 223, provide full name for DCCD.

16. Line 416. Mitophagy is already mentioned. Explain it in its first appearance.

17. Line 419. UCMD models. UCMD already defined.

18. Line 440. What is the biological function of LRPPRC?

19. Line 504. Change “If” to “Although”.

Author Response

Dear Referee,

On behalf of all authors, I kindly thank you for your effort in revising our manuscript.

During the revision process, we realized that a major issue occurred in our reference manager. This caused the loss of multiple references and editing issues.

This issue has now been solved, and all comments addressed, then we are now providing a significantly improved manuscript.

We hope that you will find the new manuscript of sufficient quality to recommend its publication.

Also, please find a response to all your comments, below.

Comment 1. Although it is understood that the main body of this review manuscript is the description of pathological involvement of PTPC, a lot of information is cramped in the Introduction and therefore it got quite long. Dividing into additional headings or subheadings based on content will provide better guidance to the readers.

Reply: According to this comment we have now included subsections in the introduction. We believed that readability is significantly improved.

Comment 2. There are too many very short paragraphs (1 or 2 sentences). Sometimes, those short paragraphs seem to be used as a sort of subheadings. If so, just put them as actual subheadings. Otherwise, incorporate them into the paragraphs above or below. For example, there should be no new paragraph for lines 109-110, and lines 111-113 can be incorporated into the paragraph below. Content-wise, the paragraph (line 130-132) doesn’t seem to belong anywhere. Either expand it or move to appropriate place. 

Reply: We apologize for presenting a manuscript with so many formatting issues. Though we cannot understand how we believe this was connected to the issue with the reference manager. We have run extensive revision of the manuscript, removed unwanted paragraphs, and, in general, ensured that no sentences appeared isolated or misplaced.

  1. There are many missing references. Some examples are: lines 130-132, lines 175, 302, 304, 320…. and Concluding Remarks.

Reply: The reference manager issue was solved, you should now find the manuscript sufficiently referenced

  1. Line 90-102 describes ATP synthase as PTP. However, studies from Walker group showed that MPT occurs without c subunit or peripheral stalk, suggesting that ATP synthase is not (a part of) the PTP (PNAS 2017, 114: 3404 and 9086). This information should be included.

Reply: The studies by Walker’s group are now discussed in the manuscript.

  1. Line 170, Na/K-ATPase, how about PMCA (plasma membrane Ca-ATPase)?

Reply: As you properly pointed out, PMCAs are indeed among all the pump whose loss of activity participates in the discussed phenotype. PMCAs are now mentioned in this section.

  1. Line 198, different sources [47], indicating the ROS sources other than mitochondria. Then, need to specify the sources. Also, a large volume of literature has shown that reperfusion increases ROS from mitochondria. More recently, Michael Murphy's group showed that the succinate accumulated during ischemia produces superoxide by reverse electron transport, although Halestrep's group refutes it. Regardless, mitochondria have been shown to be the major source of ROS during reperfusion. This information needs to be included.

Reply: we agreed that mitochondria should be the most important source during reperfusion, though there are probably overlapping pathways in ROS production. We have now expanded this section, also discussing the mentioned studies.

  1. Line 32, change “under” to “below”.

Reply: Edited as recommended

  1. Line 34, remove “the” from “the that”

Reply: Edited as recommended

  1. Line 52, the “c” in the cytochrome c should be lower case italic

Reply: Edited as recommended

  1. Line 73. Chang PTOC to PTPC

Reply: Edited as recommended

  1. Line 113, “the mechanism previously described”. Change to “the aforementioned mechanism” or “the mechanism described above”. If it really meant “previously”, which study? Need reference.

Reply: Edited as recommended and references included

  1. Line 124, “to OMM swelling and to mitochondrial Ca overload”. OMM does not swell. Change to “to mitochondrial Ca overload and swelling”

Reply: We apologize for the misprint (IMM was meant). This sentence is now edited as recommended.

  1. Line 152, change RDC to RCD. þ

Reply: Edited as recommended

  1. Line 187, should explain HKII. Hexokinase? þ

Reply: We meant HK indeed. This section is now expanded

  1. Line 223, provide full name for DCCD.

Reply: Full name is now included

  1. Line 416. Mitophagy is already mentioned. Explain it in its first appearance.

Reply: Edited as recommended

  1. Line 419. UCMD models. UCMD already defined.

Reply: This redundancy was solved as recommended

  1. Line 440. What is the biological function of LRPPRC?

Reply: We now provided a brief description of LRPPRC

  1. Line 504. Change “If” to “Although”.

Reply: Edited as recommended

Reviewer 2 Report

1. The intro is pretty choppy. It seems to be a collection of sentences that are stuck together without much flow. The organization is fine, but some consideration should be given to improving the readability.

Examples of this include...
(i) Lots of isolated sentences (highlighted below), and I don't know if this is an issue with the formatting or what happened. But this needs to be fixed.

(ii) One sentence highlighting GSK3B as promoting PTPC. Why? How does this fit with the larger introduction? And how does this relate to Bax and Bak discussion in previous paragraph or ANT-Bcl-xL discussion that follows?

2. In sequential sections on acute and degenerative conditions, the organization is handled differently. I like the subsections in the degenerative section, and a similar approach could be used for the acute section that precedes it.

2a. Related to this, the last sentence in PTPC in acute conditions section seems to relate more to renal failure, so I would think it makes sense to follow that paragraph earlier in the section.

3. While OPA1 is involved in IMM fusion, it is also important for maintaining cristae junctions that limit secretion of proteins in the inter-cristal space. I don't know whether SPG7 has an indirect effect through other proteins, but this seems like an important point to make when thinking about changes in access to distinct compartments.

4. Why not refer to cancer in naming the section on "PTPC in malignancies"? All of the content seems to deal with cancer, so you can be specific.

5. Similarly, the last section entitled "Concluding remarks" focuses more on potential therapies. I would rename this section. It is fine to end with this section, but it's not a conclusion.

6. The TWO figures look nice, but they are never refered to in the manuscript. NOT ONCE. There has to be some relationship, so please find the connections.

Specific examples of isolated sentences...

Line 61-64, unintended carage return(s)? I am not sure that any of these sentences stand alone as a paragraph. Incorporate into previous or sequential paragraph as appropriate.

Lines 109-114, again...isolated sentences. Not sure why these stand alone.

Line 132, another isolated sentence. Seems to relate back to previous paragraph.

Line 149, a blanket statement as an isolated sentence. No context and not sure what "this evidence" refers to.

Line 158, why does a separate paragraph begin? This follows directly from previous sentence.

Line 203, another isolated sentence, and a big compound sentence. Break up into separate sentences and it may stand alone (?).

Line 318. this sentence points out an interesting pattern in AD patients, but this should be expounded with examples. No citations for this broad statement.

Line 351, two opening sentences are quite general. The relationship with mitochondrial dysfunction directly follows.

Other minor points...

Line 88, typo ATN instead of ANT

I am not sure why Ischemia is abbreviated at IS. It is a single word. So an abbreviation isn't needed and may add confusion.

Some brief explanation of Ppif KO mice would help. Never state that these are CyPD KO.

ColVI genes are eluded to in Line 401, but the explanation of the abbv is provided in the next paragraph (line 411).

Author Response

Dear Referee,

On behalf of all authors, I kindly thank you for your effort in revising our manuscript.

During the revision process, we realized that a major issue occurred in our reference manager. This caused the loss of multiple references and editing issues.

This issue has now been solved, and all comments addressed, then we are now providing a significantly improved manuscript.

We hope that you will find the new manuscript of sufficient quality to recommend its publication.

Also, please find a response to all your comments, below.

Comment 1. The intro is pretty choppy. It seems to be a collection of sentences that are stuck together without much flow. The organization is fine, but some consideration should be given to improving the readability.

Examples of this include...

(i) Lots of isolated sentences (highlighted below), and I don't know if this is an issue with the formatting or what happened. But this needs to be fixed.

(ii) One sentence highlighting GSK3B as promoting PTPC. Why? How does this fit with the larger introduction? And how does this relate to Bax and Bak discussion in previous paragraph or ANT-Bcl-xL discussion that follows?

Reply: We have extensively edited the introduction (including subsections) to improve its readability.

Comment 2. In sequential sections on acute and degenerative conditions, the organization is handled differently. I like the subsections in the degenerative section, and a similar approach could be used for the acute section that precedes it.

Reply: According to your comment, we have now included subsections for the chapter “PTPC in acute conditions”

Comment 2a. Related to this, the last sentence in PTPC in acute conditions section seems to relate more to renal failure, so I would think it makes sense to follow that paragraph earlier in the section.

Reply: After this comment, we reasoned about the proper placing of AKI in the manuscript. As it showed manifestation similar to ischemia/RI, but it often occurs at nominal pO2 levels, we believed that it was better to include it in a separate subsection, to avoid confusion in the reader.

Comment 3. While OPA1 is involved in IMM fusion, it is also important for maintaining cristae junctions that limit secretion of proteins in the inter-cristal space. I don't know whether SPG7 has an indirect effect through other proteins, but this seems like an important point to make when thinking about changes in access to distinct compartments.

Reply: We have expanded this section including pointing the possible mechanism of action of OPA1 and SPG7 on PTPC activity

  1. Why not refer to cancer in naming the section on "PTPC in malignancies"? All of the content seems to deal with cancer, so you can be specific.

Reply: This chapter is now renamed “PTPC in cancer”

  1. Similarly, the last section entitled "Concluding remarks" focuses more on potential therapies. I would rename this section. It is fine to end with this section, but it's not a conclusion.

Reply: This chapter is now renamed “Potential of PTPC targeting and Concluding remarks”

Comment 6. The TWO figures look nice, but they are never refered to in the manuscript. NOT ONCE. There has to be some relationship, so please find the connections.

Reply: the authors apologize for this naive mistake. Figures are now cited in the manuscript.

Comment 7. Specific examples of isolated sentences...

Line 61-64, unintended carage return(s)? I am not sure that any of these sentences stand alone as a paragraph. Incorporate into previous or sequential paragraph as appropriate.

Lines 109-114, again...isolated sentences. Not sure why these stand alone.

Line 132, another isolated sentence. Seems to relate back to previous paragraph.

Line 149, a blanket statement as an isolated sentence. No context and not sure what "this evidence" refers to.

Line 158, why does a separate paragraph begin? This follows directly from previous sentence.

Line 203, another isolated sentence, and a big compound sentence. Break up into separate sentences and it may stand alone (?).

Line 318. this sentence points out an interesting pattern in AD patients, but this should be expounded with examples. No citations for this broad statement.

Line 351, two opening sentences are quite general. The relationship with mitochondrial dysfunction directly follows.

Reply: We apologize for presenting a manuscript with so many formatting issues. Though we cannot understand how we believe this was connected to the issue with the reference manager. All isolated sentences are now properly placed.

Comment 8 Other minor points...

Line 88, typo ATN instead of ANT

Reply: the typo is now corrected

I am not sure why Ischemia is abbreviated at IS. It is a single word. So an abbreviation isn't needed and may add confusion.

Reply: Ischemia is now spelled in full length across the manuscript.

Some brief explanation of Ppif KO mice would help. Never state that these are CyPD KO.

Reply: We have now clearly stated that Ppif gene codes for CypD and that Ppif KO animals do not express CypD

ColVI genes are eluded to in Line 401, but the explanation of the abbv is provided in the next paragraph (line 411).

Reply: Abbreviation of ColVI is now properly placed

Reviewer 3 Report

It is well written and wonderful review article.

Author Response

Dear Referee,

On behalf of all authors, I kindly thank you for your effort in revising our manuscript and for your positive evaluation.